# Combined Strategies for Improving Aflatoxin B_1_ Degradation Ability and Yield of a *Bacillus licheniformis* CotA-Laccase

**DOI:** 10.3390/ijms25126455

**Published:** 2024-06-12

**Authors:** Yanrong Liu, Limeng Liu, Zhenqian Huang, Yongpeng Guo, Yu Tang, Yanan Wang, Qiugang Ma, Lihong Zhao

**Affiliations:** 1State Key Laboratory of Animal Nutrition and Feeding, Poultry Nutrition and Feed Technology Innovation Team, College of Animal Science and Technology, China Agricultural University, Beijing 100193, China; 15110578158@163.com (Y.L.); liulimeng_9@163.com (L.L.); hzq0921@cau.edu.cn (Z.H.); m17801115235@163.com (Y.T.); wyn17600596712@163.com (Y.W.); maqiugang@cau.edu.cn (Q.M.); 2College of Animal Science and Technology, Henan Agricultural University, Zhengzhou 450046, China; 18771951786@163.com

**Keywords:** CotA-laccase, site-directed mutagenesis, signal peptide optimization, aflatoxin B_1_

## Abstract

Aflatoxin B_1_ (AFB_1_) contamination is a serious threat to nutritional safety and public health. The CotA-laccase from *Bacillus licheniformis* ANSB821 previously reported by our laboratory showed great potential to degrade AFB_1_ without redox mediators. However, the use of this CotA-laccase to remove AFB_1_ in animal feed is limited because of its low catalytic efficiency and low expression level. In order to make better use of this excellent enzyme to effectively degrade AFB_1_, twelve mutants of CotA-laccase were constructed by site-directed mutagenesis. Among these mutants, E186A and E186R showed the best degradation ability of AFB_1_, with degradation ratios of 82.2% and 91.8% within 12 h, which were 1.6- and 1.8-times higher than those of the wild-type CotA-laccase, respectively. The catalytic efficiencies (*k*_cat_/K_m_) of E186A and E186R were found to be 1.8- and 3.2-times higher, respectively, than those of the wild-type CotA-laccase. Then the expression vectors pPICZαA-N-E186A and pPICZαA-N-E186R with an optimized signal peptide were constructed and transformed into *Pichia pastoris* GS115. The optimized signal peptide improved the secretory expressions of E186A and E186R in *P. pastoris* GS115. Collectively, the current study provided ideal candidate CotA-laccase mutants for AFB_1_ detoxification in food and animal feed and a feasible protocol, which was desperately needed for the industrial production of CotA-laccases.

## 1. Introduction

Mycotoxins are a diverse, large group of fungal secondary metabolites that contaminate food and feed worldwide [1]. Aflatoxins are the most common mycotoxins produced by *Aspergillus* species, such as *A. nomius* and *A. flavus*, and pose significant hazards to the health of humans and animals [2,3]. Among aflatoxins, aflatoxin B_1_ (AFB_1_) has received special attention due to its severe hepatotoxic, teratogenic, as well as carcinogenic toxicity [4]. Furthermore, AFB_1_ contamination causes huge economic losses to food and feed production each year [5]. Therefore, it is necessary to develop efficient and environmentally friendly strategies for AFB_1_ detoxification. Several studies have reported the successful detoxification of AFB_1_ by enzymes, which provided a new perspective on reducing AFB_1_ contamination [6,7,8,9]. Laccases are oxidases that play a vital role in AFB_1_ degradation [10]. Our previous studies obtained a *Bacillus licheniformis* CotA-laccase [8], which could degrade AFB_1_ to AFQ_1_ and *epi*-AFQ_1_. AFQ_1_ is about 18-times less toxic to chicken embryo than AFB_1_ and is lacking genotoxicity [11,12]. Moreover, both AFQ_1_ and *epi*-AFQ_1_ were almost non-toxic to human liver cells [8], indicating the degradation of AFB_1_ by *Bacillus licheniformis* CotA-laccase is an effective method. This CotA-laccase has the advantage of high thermostability, low toxicity of AFB_1_ degradation products, and degrading AFB_1_ without redox mediators, which makes it have the application prospect of being developed as an animal feed additive.

Nowadays, an increasing number of researchers are employing molecular modification technology to obtain excellent mutants of enzymes that meet the requirements of industrial applications [13]. Some studies have reported that molecular modification could be used to enhance the mycotoxin degradation abilities of enzymes. Lin et al. introduced positively charged lysine mutations on the surface of the zearalenone degrading enzyme ZHD101, and obtained two mutants, D157K and E171K, with increased catalytic efficiencies under acidic conditions [14]. In the study of Ma et al., a double-mutant L110V/V429A of the deoxynivalenol degrading enzyme DADH with improved preference for deoxynivalenol was constructed based on pocket reshaping [15]. Additionally, in the study of Liu et al., a CotA-laccase mutant, Q441A, was constructed with improved AFB_1_ degradation ability [16]. Most of the reported studies on the molecular modification of CotA-laccases focus on improving its dye decolorization efficiency [13,17,18], and there are few studies about improving its mycotoxin degradation ability by molecular modification.

The expression systems for recombinant protein production include eukaryotic and prokaryotic cells [19]. Among of them, *Escherichia coli* and yeast cells are most commonly used for protein expression [20]. Compared to the *E. coli* expression system, the methylotrophic *Pichia pastoris* expression system has many advantages, such as the post-translational modification of heterologous protein and good genetic stability [21]. Moreover, the secretion expression of heterologous proteins can avoid the need for cell disruption, thereby improving the yield of proteins and reducing the cost, which in turn simplifies the downstream process [22,23]. In the *P. pastoris* expression system, the construction of an efficient expression vector is important for the expression of heterologous proteins [24]. Some studies indicated that the modification of the α-factor signal peptide in the vector could improve the expression level of recombinant protein in the *P. pastoris* expression system [25]. Consequently, it is assumed that there is potential to improve the yield of recombinant CotA-laccases in the *P. pastoris* expression system by using an optimized expression vector.

In this research, we tried to use combined strategies, which include single-point mutation and protein expression in *P. pastoris* based on an optimized vector, to increase the AFB_1_ degradation rate and yield of CotA-laccase. Two mutants with an improved AFB_1_ degradation ability were constructed by site-directed mutagenesis. Then these mutants were expressed in *P. pastoris* GS115 based on the vector with the optimized signal peptide to improve their yield. Overall, the mutant enzymes constructed in this study will be better candidates for AFB_1_ detoxification in food and feed industries.

## 2. Results and Discussion

AFB_1_ is one of the most toxic mycotoxins and poses a serious threat to humans and animals worldwide. Using CotA-laccases to degrade AFB_1_ has been proved to be environmentally friendly and safe. In order to improve the AFB_1_ degradation rate and the yield of CotA-laccase, the combined strategies, including single-point mutation and protein expression in *P. pastoris* based on the optimized vector, were used in this study.

### 2.1. Design, Expression, and Characterization of the Mutants

In our research, twelve mutants, including E186A, E186R, E186K, F205Y, R208G, K315N, G322S, A376I, L385W, I417G, P454S, and D500G, were designed for screening CotA-laccase candidates with a high AFB_1_ degradation ability according to the previous studies (Table 1). Some mutation sites in these mutants were near the catalytic center, such as 205Phe, 208Arg, 322Gly, 376Ala, 385Leu, 417Ile, and 500Asp. The mutation of these sites might show new interactions between amino acids and substrate, and thereby conducive to the combination of substrates and CotA-laccase [18,26,27,28,29]. The others were located on the surface of the protein structure, such as 186Glu, 315Lys, and 454Pro. A study indicated that the substitution of 186Glu had an influence on the activity of CotA-laccase by reshaping the binding pocket [30]. The amino acid at position 315 pointed to the interior of CotA-laccase. Changing the amino acid at this position to Asn, which was frequently found at the corresponding position of laccases in bacteria and fungi, could improve the solubility and decolorization ability of CotA-laccase [27]. The site 454Pro maintained the substrate binding pocket. The substitution of this amino acid may affect the catalytic activity of CotA-laccase [26]. We speculated these mutations would contribute to increasing the AFB_1_ degradation rate of CotA-laccase in this study.

As shown in Figure 1A, the WT and twelve mutants were all purified successfully with the molecular weight of approximately 60 kDa. The catalytic activity of CotA-laccases using ABTS or AFB_1_ as the substrate was analyzed. All of the CotA-laccases had catalytic activity toward ABTS and AFB_1_. The enzyme activity tests revealed a remarkable improvement of ABTS and AFB_1_ degradation by E186A, E186R, R208G, K315N, A376I, I417G, and D500G (Figure 1B and Appendix A). Compared with the WT, all mutated CotA-laccases showed the same trend of catalytic activity for ABTS and AFB_1_. Among all of the positive mutated CotA-laccases, E186A and E186R showed the highest AFB_1_ degradation rates for 82.2% and 91.8%, respectively. These two purified mutated CotA-laccases were chosen for the further studies. The effects of pH and temperature on the activity and stability of CotA-laccases with ABTS as the substrate are shown in Figure 2. The optimal pH of the WT and E186A were both pH 4 with ABTS as the substrate (Figure 2A). E186R revealed a slight shift toward a more acidic value with the optimal pH ranging from 4 to 3. Additionally, E186A and E186R displayed improved activity at pH 2 and 3 than the WT. The WT, E186A, and E186R were all stable at pH 6–11 (Figure 2B). The WT was most stable at pH 9, while E186A and E186R were most stable at pH 10 and 8, respectively. The pH stability of E186A and E186R had no distinct difference with that of the WT. The optimal temperatures of the WT, E186A, and E186R were 90 °C using ABTS as substrate (Figure 2C). The WT, E186A, and E186R exhibited more than 50% activity after incubation at 50 °C for 5 h (Figure 2D). All of the three enzymes maintained more than 50% activity after incubation at 70 °C for 1 h (Figure 2E). Furthermore, the thermostability of E186A at 70 °C was higher than that of the WT. E186A and E186R persisted at 60% and 40%, respectively, of the activity after incubation at 90 °C for 5 min (Figure 2F). The good thermostability of these CotA-laccases made them suitable for high-temperature pelletization in feed processing [31,32].

### 2.2. Aflatoxin B_1_ Degradation

The application of *B. licheniformis* CotA-laccases for AFB_1_ degradation has been reported previously [8,16,33]. To further optimize the AFB_1_ degradation system of E186A and E186R, the optimal pH and temperature of E186A and E186R to degrade AFB_1_ were explored. The results indicate that the AFB_1_ degradation rate of the WT is improved by single-point mutation at various pH and temperature levels (Figure 3). This might be due to the improved specificity of E186A or E186R to AFB_1_. The AFB_1_ degradation rate of E186A and E186R was highest at pH 9 (Figure 3A). Moreover, the AFB_1_ degradation rate of E186A and E186R at pH 5–9 increased compared with that of the WT. The pH of the animal gastrointestinal tract is 3–7 [34,35,36]. The improved AFB_1_ degradation ability of E186A and E186R under neutral and acidic conditions indicated that they could function better in the gastrointestinal tract of livestock As shown in Figure 3B, the optimal temperatures for E186A and E186R to degrade AFB_1_ are 80 and 70 °C, respectively. Similarly, the optimal AFB_1_ degradation temperature of CotA-laccase mutant Q441A was 70 °C [16]. The optimal temperature for CotA-laccase in *B. licheniformis* ZOM-1 to degrade AFB_1_ was 80 °C [33]. The optimal temperature of the WT, E186A, and E186R to degrade AFB_1_ in the current study was higher than that of laccases from fungi [37,38]. In addition, the degradation rate of AFB_1_ by E186R was higher than that by the WT and E186A at 30–80 °C, and the degradation rate of AFB_1_ by E186A was higher than that of the WT at 30, 70, and 80 °C. The above results indicate that E186A and E186R exhibit excellent AFB_1_ degradation abilities in a broad range of temperatures so that they are suitable as feed additives for the detoxification of AFB_1_.

The kinetic parameters of the WT, E186A, and E186R were determined on AFB_1_. The data are shown in Table 2. Despite the distance between the mutated residue and active site of the enzyme, the catalytic efficiency (*k*_cat_/K_m_) of E186A and E186R increased by 1.8 and 3.2 times in comparison with that of the WT, respectively, which was attributed to the decreased K_m_ and increased *k*_cat_. The K_m_ value of the WT to AFB_1_ decreased from 0.191 to 0.109 and 0.177 after 186Glu mutated to Ala and Arg, respectively, indicating that the enzyme affinity of E186A and E186R to AFB_1_ increased. Moreover, the *k*_cat_ of E186R was 2.9-times higher than that of the WT. A similar result was found for the E188R mutant of *Bacillus* sp. HR03 [30]. However, the *k*_cat_ of the E188A mutant of *Bacillus* sp. HR03 was slightly decreased from 20 to 18 [30].

The three-dimensional structure of CotA-laccase in this study was generated by homologous modeling on the Swiss Model. The global model quality estimate (GMQE) was 0.90, indicating that this structure was ideal (Appendix A). Then the local protein structure of the amino acid residue at site 186 and neighboring residues were analyzed. Glu186 was located on the surface of this enzyme and at a certain distance from the catalytic center (Figure 4A). The change in hydrogen bonds at the local region of the mutants is shown in Figure 4B–D. In the WT, Glu186 formed hydrogen bonds with Lys340 and Lys343 (Figure 4B). When Glu was substituted by Ala, no hydrogen bond formed between the Ala186 with neighboring residues (Figure 4C). The mutation of Glu186 to Arg186 also showed the same result as the mutant E186A (Figure 4D). The disruption of old hydrogen bonds might affect the shape and size of the binding pocket and make the substrate easily enter the active center of CotA-laccase. Moreover, the mutation of Glu186 might eliminate the juxtaposition of phenolic rings between adjacent residues, thereby reducing steric hindrance and facilitating the binding of the substrate to the catalytic center [30].

### 2.3. Secretory Expression of CotA-Laccases in Pichia Pastoris GS115 with the Optimized Vector

The practical, industrial application of enzyme preparation by heterologous expression in *E. coli* has some disadvantages, such as the low yield of enzymes and high cost due to cell disruption [22,23]. Therefore, the best mutated genes, E186A and E186R, obtained in the previous step were transformed into *P. pastoris* for secretory expression. It has been reported that adding the spacer peptide “EEAEAEAEPK” to the end of the α-factor signal peptide was able to increase the yield of insulin precursor secretory expressed by *P. pastoris* [39], so we speculated that the secretory expression of CotA-laccases might be improved using the vector containing this spacer peptide, thereby enhancing the AFB_1_ degradation ability of the *P. pastoris* fermentation supernatant. The genes of CotA-laccases were cloned into the vector pPICZαA-N, which was optimized by adding the spacer “EEAEAEAEPK” to the end of the α-factor signal peptide of the vector pPICZαA, and were transformed into *P. pastoris* GS115 (GS115-pPICZαA-N-WT/E186A/E186R, Figure 5A). *P. pastoris* GS115 containing pPICZαA connected with the gene of the WT (GS115-pPICZαA-WT) was used as the control to determine the AFB_1_ degradation ability of the fermentation supernatant. The CotA-laccases were successfully secretory expressed by *P. pastoris* GS115 (Appendix A). The CotA-laccases activity of the fermentation supernatant of *P. pastoris* reached the highest value after 7 d of cultivation (Figure 5B). In the study of Song et al., the laccase Lac2 from *Pleurotus pulmonarius* was expressed in *P. pastoris*, and the maximum laccase activity of the crude enzyme solution appeared at 14 d of cultivation [37]. The CotA-laccase activity of the fermentation supernatant of GS115-pPICZαA-N-E186R was always higher than the others during the 10 d of induction, reaching a maximum of 447.2 U/L. The AFB_1_ degradation rate of the fermentation supernatant was determined after induction for 7 d. The degradation rate of AFB_1_ by the fermentation supernatants of GS115-pPICZαA-N-WT, GS115-pPICZαA-N-E186A, and GS115-pPICZαA-N-E186R were higher than that of GS115-pPICZαA-WT, where the highest AFB_1_ degradation rate of 45.8% was achieved for the fermentation supernatant of GS115-pPICZαA-N-E186R (Figure 5C). This result indicates that the vector with the optimized signal peptide increases the secretory expression of CotA-laccases, which is consistent with the result of insulin precursor secretory expression in *P. pastoris* by Kjeldsen et al. [39]. GS115-pPICZαA-N-E186R may be a good candidate engineering bacterium to be used in practical industrial applications to produce E186R with a high AFB_1_ detoxification ability.

## 3. Materials and Methods

### 3.1. Materials

2,2′-Azino-bis (3-ethylbenzothiazoline-6-sulfonic acid) (ABTS) and isopropyl-β-D-thiogalactopyranoside (IPTG) were purchased from Biotopped (Beijing, China). TIANprep Mini Plasmid Kit, Fast Site-Directed Mutagenesis Kit, TIANgel Purification Kit, and TIANamp Yeast DNA Kit were obtained from Tiangen (Beijing, China). The BCA Protein Quantitative Analysis Kit was purchased from Aidlab (Beijing, China). AFB_1_ was purchased from Pribolab (Qingdao, China). Ampicillin and zeocin were obtained from Solarbio (Beijing, China). Restriction enzymes were purchased from Takara (Beijing, China). All other reagents were of analytical grade or higher and obtained from Merck (Darmstadt, Germany). The strain *B. licheniformis* ANSB821 (NCBI accession number of 16S rDNA: MN075270) was isolated and preserved in our laboratory [8]. The engineering bacterium DE3-pET31b-CotA was previously constructed in our laboratory [8]. pET-31b (Takara, China) was used to clone the genes of the wild-type CotA-laccase (WT) and its mutants for subcloning in *E. coli* DH5α cells (Takara, China) and protein expression in *E. coli* BL21 (DE3) cells (Takara, China). The pPICZαA vector was purchased from Takara (Beijing, China). pPICZαA-N was a vector optimized by adding a spacer, “EEAEAEAEPK”, to the end of the α-factor signal peptide of the vector pPICZαA, and was used to clone the genes mentioned above for the secretion expression of heterologous proteins in *P. pastoris* GS115 cells (Takara, China) [38]. This optimized vector, pPICZαA-N, was supplied by Prof. Yunhe Cao from the College of Animal Science and Technology in China Agricultural University. *E. coli* DH5α cells and *E. coli* BL21 cells were grown in Luria–Bertani (LB) medium at 37 °C. *P. pastoris* GS115 cells were grown in yeast extract–peptone–dextrose (YPD) medium, buffered minimal methanol (BMMY) medium, or buffered glycerol-complex (BMGY) medium at 28 °C.

### 3.2. Bioinformatics Analysis

All protein sequences were searched from the National Center for Biotechnology Information (NCBI, https://www.ncbi.nlm.nih.gov, accessed on 15 March 2023). Homology modeling was performed with the Swiss-model server (http://swissmodel.expasy.org/, accessed on 16 March 2023). Among the known three-dimensional CotA models, CotA-laccase in the current study shares the maximum protein sequence identity (65.29%) with the spore coat protein A CotA-laccase (PDB code: 4A67). The visualization of the protein structure was achieved by UCSF Chimera 1.11.2. The local structure of CotA-laccases was analyzed using the ProteinTools server (proteintools.uni-bayreuth.de).

### 3.3. Site-Directed Mutagenesis

The mutants were constructed with the Fast Site-Directed Mutagenesis Kit, as described by Liu et al. [16]. The primers are listed in Appendix A. PCR products were digested with DpnI to remove the methylated template and transformed into *E. coli* DH5α cells. Then, the plasmids were extracted from *E. coli* DH5α and the plasmids containing correct mutations were transformed into *E. coli* BL21 cells for heterologous protein expression.

### 3.4. Expression and Purification in Escherichia coli BL21

*E. coli* BL21 cells harboring the genes of the WT or mutants were grown in LB medium containing 100 mg/mL of ampicillin for 12 h. The 5 mL fermentation liquid was inoculated into fresh 500 mL LB medium and incubated at 37 °C. A total of 0.1 mM of IPTG and 2 mM of CuSO_4_ were added to the medium when the optical density at 600 nm reached 0.6. After incubating for 20 h at 16 °C, the cells were harvested by centrifugation (8000× *g*, 4 °C, 15 min) and the pellets were resuspended in binding buffer (50 mM of sodium phosphate, 500 mM of NaCl, 5 mM of imidazole, and pH 7.4). The cells were disrupted by sonification and then centrifugated to remove the pellets. The purification of the WT and mutants was performed by following procedures from our previous study [8]. Then, the purified enzymes were analyzed by SDS-PAGE and the concentration of proteins was determined with the BCA Protein Quantitative Analysis Kit.

### 3.5. Enzyme Assay and Characterization

The activity of all CotA-laccases was measured using AFB_1_ and ABTS as substrates [16]. The AFB_1_ degradation rates of the CotA-laccases were determined in 500 μL of sodium phosphate buffer (100 mM, pH 8) containing 20 μg mL^−1^ of the WT or the mutants and 2 μg mL^−1^ of AFB_1_. The control was prepared in the absence of CotA-laccases. The reaction lasted for 12 h at 37 °C. High-performance liquid chromatography (HPLC) was used to determine the AFB_1_ content in the reaction system. The catalytic activity of the CotA-laccases with ABTS as the substrate was determined in 1 mL of sodium citrate buffer (100 mM, pH 4) supplemented with 2 μg mL^−1^ of CotA-laccases and 1 mM of ABTS. The reaction lasted for 3 min at 37 °C and then the absorbance of the reaction solution at 420 nm was measured to calculate the catalytic activity of CotA-laccases to ABTS. One unit was defined as how many CotA-laccases were required to oxidize 1 µmol of ABTS every minute. The effects of pH and temperature on the activity and stability of CotA-laccases were measured as previously described by Liu et al. [16]. All experiments were replicated three times.

### 3.6. Aflatoxin B_1_ Oxidase Properties

The enzymatic treatment of AFB_1_ was performed in 0.5 mL of different buffers (100 mM of sodium citrate for pHs 5 and 6, 100 mM of sodium phosphate for pHs 7 and 8, and 100 mM of glycine-NaOH for pH 9) containing 20 µg mL^−1^ of CotA-laccase and 1 µg mL^−1^ of AFB_1_. The effect of pH and temperature on AFB_1_ oxidation by CotA-laccases was measured as described by Liu et al. [16]. The reaction system without the WT and mutants was used as the control. The catalytic rate constant (*k*_cat_) and Michaelis–Menten constant (K_m_) were measured at 37 °C and pH 8 with a concentration of AFB_1_ ranging from 1 to 100 µg mL^−1^ using the Michaelis–Menten equation. All the experiments were performed in triplicate.

### 3.7. Codon Optimization and Secretory Expression of CotA-Laccases in P. pastoris GS115

The codon-optimized genes of CotA-laccases were designed according to *P. pastoris* codon usage. pPICZαA containing the optimized sequence of the WT and pPICZαA-N containing the optimized sequences of the WT, E186A, or E186R were synthesized by Genewiz (Beijing, China). The constructed plasmids were confirmed by DNA sequencing in Sangon (Beijing, China). Then, they were transformed into *P. pastoris* GS115 according to Song et al. [37]. A total of 500 µL of positive transformants were cultured in 50 mL of BMGY medium at 28 °C and 200 rpm until the optical density at 600 nm reached 2–6. The cultures were centrifuged (5000× *g*, 4 °C, 10 min), and the cells were resuspended using 50 mL of BMMY medium. The flasks were cultivated at 28 °C and 200 rpm for 10 days, and 1% (*v*/*v*, final concentration) methanol was added every 24 h. The catalytic activity of the fermentation supernatant to ABTS and AFB_1_ was determined every 24 h.

### 3.8. Catalytic Activity of P. pastoris GS115 Fermentation Supernatant to ABTS and Aflatoxin B_1_

The catalytic activity of the *P. pastoris* GS115 fermentation supernatant to ABTS was measured by adding 20 µL 50 mM of ABTS (final concentration of 1 mM) to 1 mL of fermentation supernatant, and the pH was adjusted to 4. The reaction lasted for 3 min at 37 °C. The catalytic activity of the *P. pastoris* GS115 fermentation supernatant to AFB_1_ was measured by adding 10 µL 50 µg mL^−1^ of AFB_1_ (final concentration of 1 µg mL^−1^) to 500 µL of fermentation supernatant, and the pH of the mixture was adjusted to 8. The reaction lasted for 12 h at 37 °C. All the experiments were carried out at least in triplicate.

### 3.9. Detection of Aflatoxin B_1_ by HPLC

The AFB_1_ degradation rate was determined by HPLC. The detection method referred to that of Guo et al. [8]. Simply put, the same volume of methanol was added to the samples. Then, the samples were centrifugated and filtered through a 0.22 µm filter before HPLC analysis. The HPLC system (Shimadzu LC-10 AT, Shimadzu, Tokyo, Japan) connected to a Diamonsil^®^ C18 reverse-phase column (5 µm, 4.6 × 100 mm) was used for AFB_1_ analysis. The mobile phase was methanol:water (45:55 by volume) and the flow rate was 1 mL/min. The excitation and emission wavelengths were 360 and 440 nm, respectively. The injection volume was 50 µL. The AFB_1_ degradation rate was calculated with the following formula: AFB_1_ degradation (%) = [(Control − Sample)/Control] × 100.

## 4. Conclusions

In this study, two mutants, E186A and E186R, with an enhanced AFB_1_ degradation ability were obtained by expression in *E. coli* BL21. The AFB_1_ degradation rate reached 82.2% and 91.8% at pH 8 and 37 °C for 12 h. The catalytic efficiency of E186A and E186R increased by 1.8 and 3.2 times in comparison with that of the WT, respectively. Moreover, the genes of E186A and E186R were cloned into the vector pPICZαA-N with the optimized α-factor signal peptide, and were transformed into *P. pastoris* GS115. The optimized secretory expressions of E186A and E186R improved the AFB_1_ degradation ability of the fermentation supernatant. In summary, the AFB_1_ degradation ability and yield of CotA-laccase were improved through the combination of site-directed mutagenesis and secretory expression in *P. pastoris* GS115 based on the optimized vector. Our study provided ideal-candidate CotA-laccase mutants for AFB_1_ detoxification in food and animal feed and a feasible protocol, which was desperately needed for the industrial production of CotA-laccases.

## Figures and Tables

**Figure 1 ijms-25-06455-f001:**
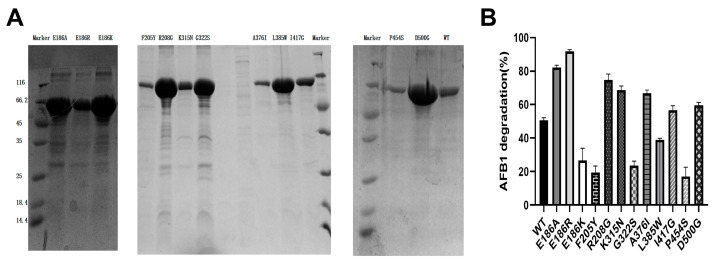
SDS-PAGE analysis of constructed wild-type CotA-laccase (WT) and the mutants (**A**) and the AFB_1_ degradation rate of the WT and mutants (**B**).

**Figure 2 ijms-25-06455-f002:**
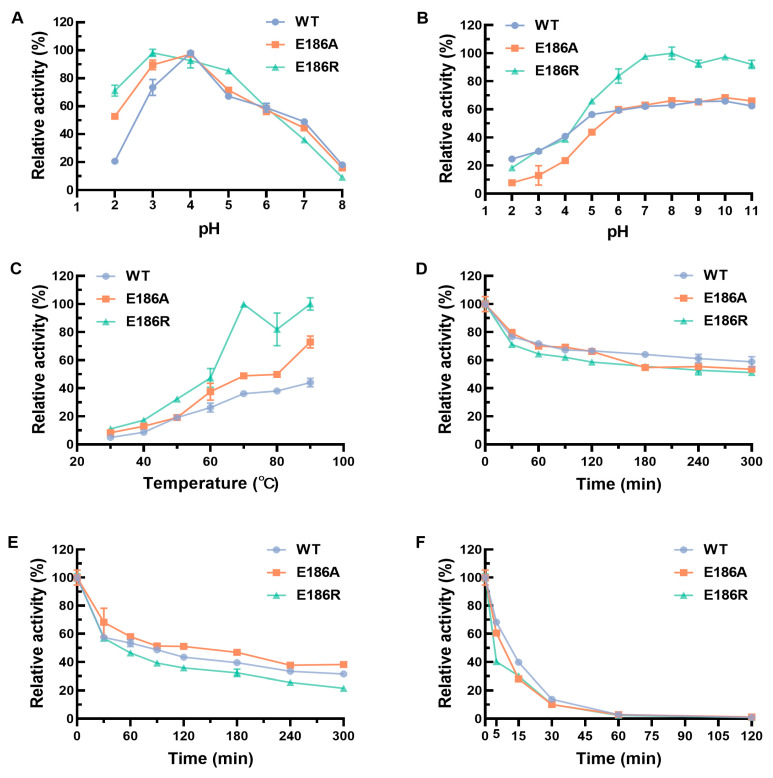
Effects of pH and temperature on the activity and stability of the purified WT and the mutants using ABTS as the substrate. (**A**) The effect of pH on CotA-laccase activity was conducted at pH 2–8; (**B**) the effect of pH on the stability of CotA-laccases was conducted by pre-incubating the purified CotA-laccases at 4 °C and pH 2–11 for 12 h; (**C**) the effect of temperature on CotA-laccase activity was conducted at 30–90 °C; the effect of temperature on the stability of CotA-laccases was conducted by pre-incubating the purified CotA-laccases at 50 °C (**D**), 70 °C (**E**), and 90 °C (**F**).

**Figure 3 ijms-25-06455-f003:**
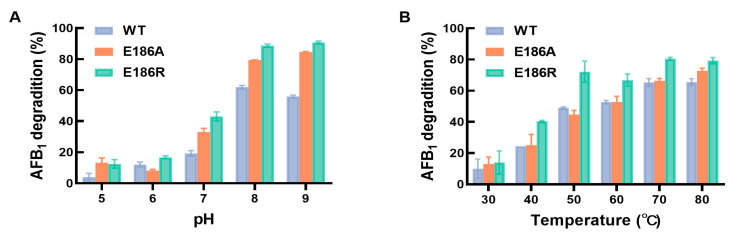
Effects of pH (**A**) and temperature (**B**) on the activity of the purified WT and the mutants using AFB_1_ as the substrate.

**Figure 4 ijms-25-06455-f004:**
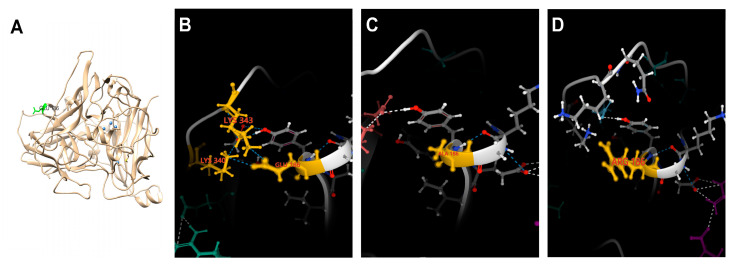
Homology model of WT, E186A, and E186R. (**A**) Glu186 is shown in green. The copper atoms are highlighted in blue; the hydrogen bonds between the side chain of the residue at site 186 and the neighboring residues in WT (**B**), E186A (**C**), and E186R (**D**) are shown as blue dashes. Amino acid residues Glu186, Lys340, Lys343 in (**B**), Ala186 in (**C**), and Arg186 in (**D**) are shown in yellow.

**Figure 5 ijms-25-06455-f005:**
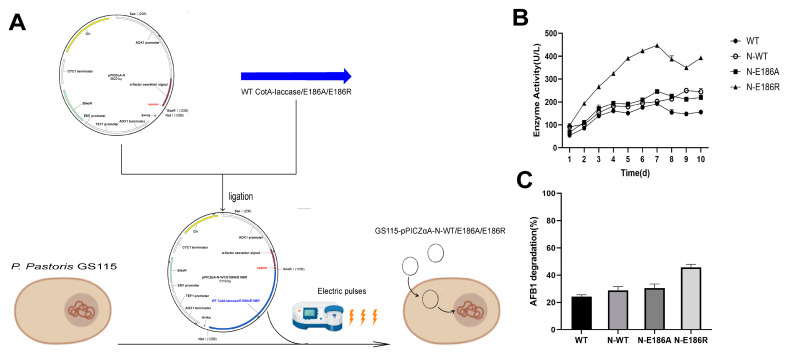
Secretory expression of CotA-laccases in Pichia pastoris GS115 with the optimized vector. (**A**) The construction process of engineering yeast GS115-pPICZαA-N-WT/E186A/E186R. (**B**) The CotA-laccase activity of the different engineering yeasts’ fermentation supernatant using ABTS as the substrate. (**C**) The AFB_1_ degradation rate of the different engineering yeasts’ fermentation supernatant.

**Table 1 ijms-25-06455-t001:** Molecular mutagenesis of CotA-laccase from *Bacillus licheniformis* ANSB821 referring to the previous studies.

Mutants Constructed in This Study	Amino Acid Changes in the Reference	Source of CotA-Laccases	Improvements of Mutants in the Reference
E186A	E188A [30]	*Bacillus* sp.	*k*_cat_/K_m_ increased from 5.0 to 5.1 s^−1^ µg^−1^ mL using SGZ ^1^ as the substrate; thermostability improved
E186R	E188R [30]	*Bacillus* sp.	*k*_cat_/K_m_ increased from 5.0 to 20.0 s^−1^ µg^−1^ mL using SGZ as the substrate; thermostability improved
E186K	E188K [30]	*Bacillus* sp.	Thermostability improved
F205Y	F207Y [26]	*B. subtilis*	*k*_cat_/K_m_ increased from 0.26 to 0.29 s^−1^ µM^−1^ using ABTS ^2^ as the substrate
R208G	S208G [18]	*B. pumilus*	*k*_cat_/K_m_ increased from 167.05 to 223.60 s^−1^ mM^−1^ using ABTS as the substrate; dye decolorization rate increased
K315N	K316N [27]	*B. licheniformis*	Dye decolorization rate increased; protein yield improved
G322S	G323S [28]	*B. pumilus*	*k*_cat_/K_m_ increased from 154.36 to 177.44 s^−1^ mM^−1^ using ABTS as the substrate; dye decolorization rate increased
A376I	T377I [28]	*B. pumilus*	*k*_cat_/K_m_ increased from 154.36 to 194.68 s^−1^ mM^−1^ using ABTS as the substrate; dye decolorization rate increased
L385W	L386W [29]	*B. subtilis*	Specificity for ABTS increased; thermostability improved
I417G	T418G [28]	*B. pumilus*	*k*_cat_/K_m_ increased from 154.36 to 203.35 s^−1^ mM^−1^ using ABTS as the substrate; dye decolorization rate increased
P454S	P455S [26]	*B. subtilis*	*k*_cat_/K_m_ increased from 0.26 to 0.83 s^−1^ µM^−1^ using ABTS as the substrate
D500G	D500G [27]	*B. licheniformis*	Dye decolorization rate increased; protein yield improved

^1^ SGZ is syringaldazine. ^2^ ABTS is 2,2′-azino-bis (3-ethylbenzothiazoline-6-sulfonic acid).

**Table 2 ijms-25-06455-t002:** Kinetic parameters for the wild-type CotA (WT), E186A, and E186R using AFB_1_ as the substrate.

CotA-Laccase	K_m_ (mM)	*k*_cat_ (s^−1^)	*k*_cat_/K_m_ (s^−1^ mM^−1^)
WT	0.191	0.072	0.377
E186A	0.109	0.073	0.670
E186R	0.177	0.211	1.192

## Data Availability

Data are contained within the article and Appendix A.

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
