# Peer review of "Combined Strategies for Improving Aflatoxin B1 Degradation Ability and Yield of a Bacillus licheniformis CotA-Laccase"

_ijms, 2024, doi:10.3390/ijms25126455_

Round 1

Reviewer 1 Report

Comments and Suggestions for Authors

The submission by Liu et al. is a continuation of their previously published research effort (https://doi.org/10.1016%2Fj.heliyon.2023.e22388). It is well written, but I have some aspects that need to be improved or clarified.

L32-34: the statement ‘‘….can act as a potent virulence factor’’ is not clear to me. What do you mean by this?

L38-39: You need to cite more studies!; Several studies can’t be a single study.

L41-42: AFQ1 is about 18 times less toxic to chicken embryo than AFB1, and it is approximately eighty-three times less mutagenic than AFB1. Please acknowledge these facts (with appropriate citations) to make it clearer to the readers, otherwise breaking aflatoxins into aflatoxins may not make sense to your audience in the present state. Also take care that although such AFs are less toxic, most of them are still carcinogenic and potent frameshift mutagens.

L65: Please expand P. pastoris at first use in the manuscript.

L193-195: Supporting references required.

L199; Delete the first ‘‘that’’

In the methods section, make references to the sources of the methods used.

L348: Description of the supplementary materials should be provided as per journal guidelines

Comments on the Quality of English Language

Minor improvements required

Author Response

L32-34: the statement "....can act as a potent virulence factor" is not clear to me. What do you mean by this?

A: Accepted. What we want to say is that aflatoxins do great harm to human and animal health. We realized that it was inappropriate to describe aflatoxins as virulence factor. The statement "....can act as a potent virulence factor" has been corrected to "....pose significant hazards to the health of humans and animals" (Highlighted, L33-34).

L38-39: You need to cite more studies!; Several studies can't be a single study.

A: Thanks for the comment. The relevant references have been added (Highlighted, L40, L380-388).

L41-42: AFQ1 is about 18 times less toxic to chicken embryo than AFB1, and it is approximately eighty-three times less mutagenic than AFB1. Please acknowledge these facts (with appropriate citations) to make it clearer to the readers, otherwise breaking aflatoxins into aflatoxins may not make sense to your audience in the present state. Also take care that although such AFs are less toxic, most of them are still carcinogenic and potent frameshift mutagens.

A: Accepted. The lower toxicity of AFQ1 and epi-AFQ1 have been explained to make the readers understand that the degradation of AFB1 by Bacillus licheniformis CotA-laccase is an effective way. In addition, the appropriate citations have been added (Highlighted, L42-45, L391-395).

L65: Please expand P. pastoris at first use in the manuscript.

A: Thank you for the comment. The manuscript has been checked to ensure that Pichia pastoris is properly formatted when it first appeared in the abstract and the text (Highlighted, L23-24, L65).

L193-195: Supporting references required.

A: Accepted. The supported reference has been added (Highlighted, L69, L199, L415-416).

L199: Delete the first "that".

A: Thanks for the comments. The first "that" in this sentence has been deleted (Highlighted, L203).

In the methods section, make references to the sources of the methods used.

A: Accepted. References have been added in the methods section (Highlighted, L267-268, L286).

L348: Description of the supplementary materials should be provided as per journal guidelines.

A: Accepted. The description of the supplementary materials has been provided as the journal guidelines (Highlighted, L353-357).

Reviewer 2 Report

Comments and Suggestions for Authors

Reduction of AFB contamination in food can be obtained by enzymatic degradation of the mycotoxin. In the present study the authors designed 12 mutants of various microorganisms of which two showed higher CotA-laccase activity. These two enzymes were further evaluated to identify opimal pH, temperature und enzyme kinetics.

The work is interesting and possibly of practical relevance.

Some smaller points should be clarified:

·      Introduction, line 39: reducing AFB1 toxicity. Better: reducing AFB contamination.

·       Please clarify the Enzyme assays. Are the mutants of the microorganisms used or the purified enzymes?

·       It is confusing that the terms mutants and enzymes are not clearly differentiated. If enzyme activity is meant the term laccase or enzyme should be used, not mutant.

Comments on the Quality of English Language

No comments. English is fine.

Author Response

Introduction, line 39: reducing AFB1 toxicity. Better: reducing AFB contamination.

A: Accepted. "reducing AFB1 toxicity" has been replaced with "reducing AFB1 contamination" (Highlighted, L39-40).

Please clarify the Enzyme assays. Are the mutants of the microorganisms used or the purified enzymes?

A: We would like to thank you for providing helpful comments on our manuscript. We are sorry that the enzyme assays is not clear. The purified enzymes were used for the enzyme assays after the wild-type CotA-laccase, E186A and E186R were heterologous expressed in E. coli. We have made modification to Enzyme assay and characterization in the Materials and Methods (Highlighted, L285, L286, L290-292).

It is confusing that the terms mutants and enzymes are not clearly differentiated. If enzyme activity is meant the term laccase or enzyme should be used, not mutant.

A: Accepted. The inappropriate term mutant in the manuscript has been changed (Highlighted, L109-118, L125, L130, L144-145, L148-150, L152, L157, L167, L285, L286, L290-292).